# Model-based estimation of left ventricular pressure and myocardial work in aortic stenosis

Kimi P. Owashi☯, Arnaud Hubert☯, Elena Galli, Erwan Donal, Alfredo I. Hernández*, Virginie Le Rolle

Univ Rennes, Inserm, LTSI - UMR 1099, Rennes, France

☯ These authors contributed equally to this work.
* Alfredo.Hernandez@inserm.fr

**Data Availability Statement:** All relevant data are within the manuscript and its Supporting Information files.

## Abstract

This paper proposes a model-based estimation of left ventricular (LV) pressure for the evaluation of constructive and wasted myocardial work of patients with aortic stenosis (AS). A model of the cardiovascular system is proposed, including descriptions of *i)* cardiac electrical activity, *ii)* elastance-based cardiac cavities, *iii)* systemic and pulmonary circulations and *iv)* heart valves. After a sensitivity analysis of model parameters, an identification strategy was implemented using a Monte-Carlo cross-validation approach. Parameter identification procedure consists in two steps for the estimation of LV pressures: *step 1)* from invasive, intraventricular measurements and *step 2)* from non-invasive data. The proposed approach was validated on data obtained from 12 patients with AS. The total relative errors between estimated and measured pressures were on average 11.9% and 12.27% and mean $R^2$ were equal to 0.96 and 0.91, respectively for steps 1 and 2 of parameter identification strategy. Using LV pressures obtained from non-invasive measurements (step 2) and patient-specific simulations, Global Constructive (GCW), Wasted (GWW) myocardial Work and Global Work Efficiency (GWE) parameters were calculated. Correlations between measures and model-based estimations were 0.88, 0.80, 0.91 respectively for GCW, GWW and GWE. The main contributions concern the proposal of the parameter identification procedure, applied on an integrated cardiovascular model, able to reproduce LV pressure specifically to each AS patient, by non-invasive procedures, as well as a new method for the non-invasive estimation of constructive, wasted myocardial work and work efficiency in AS.

## Introduction

Aortic stenosis (AS) is characterised by a narrowing of the aortic valve opening, which induces a left ventricular (LV) pressure overload. The development of LV hypertrophy in AS is accompanied by coronary microcirculatory dysfunction [1] that may gradually affect systolic and diastolic function [2]. LV ejection fraction (LVEF) is used routinely to assess LV systolic function and is an important parameter for prognosis stratification [3]. However, LVEF depends not

**Funding:** This work was supported by the French National Research Agency (ANR) (ANR-16-CE19-0008-01) (project MAESTRo) and the French Brittany council (ADvICE project).

**Competing interests:** The authors have declared that no competing interests exist.

only upon the contractility of LV, but also on loading conditions. In fact, ejection fraction may appear to be preserved despite underlying reduced contractility The characterisation of myocardial dysfunction is of primary importance to identify patients with reduced contractility. Speckle-tracking echocardiography (STE) assessment of myocardial strain usually provides a better quantification of systolic function than global LVEF [4]. Although strain echocardiography can provide prognostic information in patients with AS [5], the shortening indices, calculated from cardiac strains, do not reflect myocardial work or oxygen demand. As opposed to the normal LV, where all segments contract almost synchronously and myocardial energy is used effectively, regional dysfunction, that could be induced by myocardial fibrosis [6], could bring a significant loss of efficient work. For instance, the impairment of myocardial diastolic and systolic function, due to fibrosis [7], have shown to induce significant mechanical dispersion in patients with severe AS [8].

Recently, Russell et al [9, 10] have proposed a non-invasive method for LV work analysis, which is based upon an estimated LV pressure curve. As strain is largely influenced by LV afterload [11], model-based myocardial work might be a robust complementary tool, taking into account AS severity and arterial pressures values. In previous works of our team, we have shown that the non-invasive estimation of global myocardial work, when using an LV pressure curve estimation as proposed in [9], is correlated with that obtained when using the observed invasive LV pressure curve, in the context of cardiac resynchronization therapy [12]. However, the accuracy of estimated LV pressure has never been evaluated in the case of aortic stenosis, where high pressure gradients could be observed between LV and the aorta [13]. The experimental observation of LV pressure is notably difficult to perform clinically because it requires an invasive, intraventricular measurement. As a consequence, it is necessary to propose novel tools to assess non-invasive LV pressure and to calculate myocardial work in the case of AS.

The first objective of this paper was to propose a model-based estimation of LV pressure in the case of AS. Previous works [14, 15] has already shown that lumped-parameter models of ventricular-vascular coupling are able to provide a good agreement between the estimated and the measured left ventricular and aortic pressure waveforms. Based on these papers and previous works of our team [16–18], we proposed a model-based approach, including a multiformalism model of the cardiovascular system and a parameters identification strategy using a Monte-Carlo cross-validation method, in order to: 1) estimate LV pressure waveform from experimental LV pressure curve, systolic and diastolic aortic pressure values, 2) assess LV pressure waveform from only systolic and diastolic aortic pressure values.

The second objective of the paper was to propose a novel tool to estimate myocardial work in AS. Work indices, as proposed in [9, 10] and validated in [12], were calculated from non-invasive model-based LV pressure and compared with indices evaluated from experimental signals. This article does not claim to validate the estimation of myocardial work in a cohort of AS patients but it aims to propose an original approach for the assessment of work indices based on computational modelling. The paper is organised as follows: in Section 2, the experimental protocol and data under study are presented, the computational model is described and the identification method is explained. In Section 3, the results of applying the described methods are presented and discussed. Discussions are finally specified in Section 4.

# 1 Materials and methods

## 1.1 Experimental data

**1.1.1 Study population.** We prospectively included 12 adults (≥18 years old) with severe (aortic valve area (AVA) ≤ 1cm$^2$, n = 11) and moderate (n = 1) aortic stenosis who underwent a coronary angiography with left heart catheterization. Table 1 summarizes patients' clinical

**Table 1. Patients' clinical characteristics.**

|  | Age<br>*years old* | Male sex<br>*n (%)* | BSA<br>(body surface area) | NYHA class<br>II/III, *n* |
|---|---|---|---|---|
| Patients (n = 12) | 78.16 ± 5.50 | 7 (58.3%) | 1.75 ± 0.10 | 8/4 |

characteristics. We excluded patients with concomitant significant aortic regurgitation and mitral stenosis. The study was carried out in accordance with the principles outlined in the Declaration of Helsinki on research in human subjects and received specific ethical approval from of the local Medical Ethics Committee (Person Protection Committee West V—CPP Ouest V, authorization number: 2014-A01331-456). All patients were informed and a written consent was obtained.

**1.1.2 Echocardiography.** All patients underwent a standard Trans-Thoracic Echocardiography (TTE) using a Vivid S6, E7 or E9 ultrasound system (General Electric Healthcare, Horten, Norway). Images were recorded on a remote station for off-line analysis by dedicated software (EchoPAC PC, version BT 202, General Electric Healthcare, Horten, Norway). The analysis of aortic and mitral valve events during a complete TTE loop [mitral valve closure (MVC), aortic valve opening (AVO), aortic valve closure (AVC), mitral valve opening (MVO)] was performed in apical long-axis view and individual valvular events were manually segmented. Standard STE analysis was applied in order to extract regional myocardial strain curves. Also aortic stenosis analysis was performed to estimate the AVA ($cm^2$).

**1.1.3 Invasive experimental pressure.** The left heart catheterization (LHC) was performed via a retrograde access from the radial artery with a 5 French Judkin R4 catheter (ICU Medical, San Clemente, CA, USA) placed at the mid LV cavity using fluoroscopic screening. Before coronary angiography, transducers were calibrated, with a 0-level set at the mid-axillary line. In a second time, catheter was placed in the thoracic ascendant aorta to measure aortic pressure. The experimental invasive data set includes the measured ventricular pressure $P_{LV}^{exp}$, the systolic $P_{ao,sys}^{exp}$ and diastolic $P_{ao,dias}^{exp}$ arterial pressures.

## 1.2 Computational model

Four main sub-models were created and coupled: *i)* cardiac electrical system, *ii)* elastance-based cardiac cavities, *iii)* systemic and pulmonary circulations and *iv)* heart valves. The first three submodels are strongly based on our previous works [16–21]. The model of the heart valves was adapted from [22].

**1.2.1 Cardiac electrical system.** The proposed model of the cardiac electrical activity, is based on a set of coupled automata [19, 20] (Fig 1). Each automaton represents the electrical activation state of a given myocardial tissue, covering the main electrophysiological activation periods: slow diastolic depolarisation (SDD), upstroke depolarization (UDP), absolute refractory (ARP) and relative refractory (RRP). Briefly, the state of the cellular automata cycles through these four stages, sending an output stimulation signal to neighboring cells when a given cell is activated (end of UDP phase).

The whole simplified model consists of seven automata representing: the sinoatrial node (NSA), right and left atria (RA and LA), the atrioventricular node (NAV), upper bundle of His (UH) and both ventricles (RV and LV). The electrical activation of the automata is used to synthesize an electrocardiogram (ECG), from which the QRS peak was extracted to synchronize the experimental and simulated signals.

**1.2.2 Elastance-based cardiac cavities.** Although the literature offers a wide range of cardiovascular models, elastance-based models offer a good compromise between complexity and

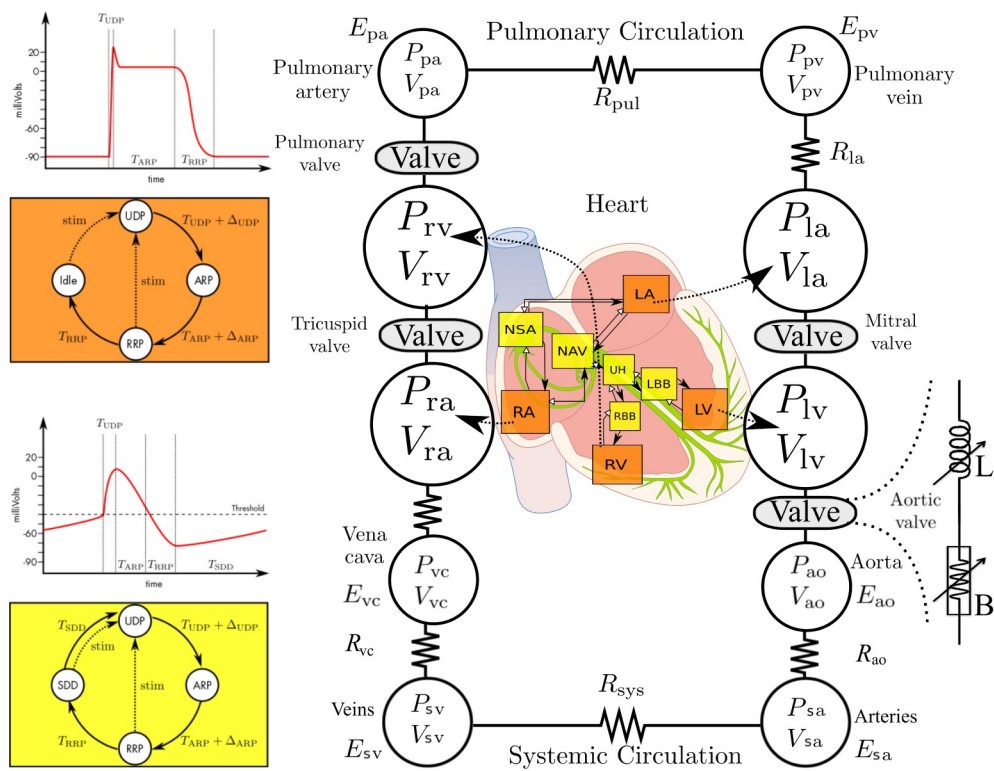

**Fig 1.** *Left panel*: State diagram of the cellular automata that represent nodal cells (yellow, botton) and myocardial cells (orange, top) and diagrams showing the correspondence of the automata's transition parameters with the myocardial action potential dynamics. *Right panel*: Closed-loop model of the cardiovascular system. E: elastance; R: resistance; P: pressure; V: volume; pul: pulmonary; sys: systemic; pv: pulmonary vein; pa: pulmonary artery; ao: aorta; sa: systemic artery; sv: systemic veins; vc: vena cava; LA: left atrium; LV: left ventricle; RA: right atrium; RV: right ventricle. In the middle, a representation of the cardiac electrical system. On the right, a representation of the heart valve model.

number of parameters [16–18, 23]. Ventricle pressures are represented by a combination of the end-systolic (*es*) and end-diastolic (*ed*) pressure-volume relationships [24]:

$$P_{es,lv}(V, t) = E_{es,lv}(V(t) - V_{d,lv}), \qquad (1)$$

$$P_{ed,lv}(V, t) = P_{0,lv}(e^{\lambda_{lv}(V(t) - V_{0,lv})} - 1) \qquad (2)$$

In Eq 1, systolic pressure $P_{es,lv}$ is defined as a linear relationship with the ventricular volume $V$, determined by the systolic elastance $E_{es,lv}$ and the volume intercept $V_{d,lv}$. Eq 2 also describes the nonlinear diastolic pressure defined by a gradient $P_{0,lv}$, curvature $\lambda_{lv}$ and volume intercept $V_{0,lv}$. The instantaneous pressure of the ventricle is then calculated as:

$$P_{lv}(V, t) = e_{lv}(t)P_{es,lv}(V, t) + (1 - e_{lv}(t))P_{ed,lv}(V, t) \qquad (3)$$

where $e_{lv}(t)$ is the driver function that controls time-variant elastance. In this work we have

selected a "double Hill" driver function [25] that best fits our observed data:

$$e_{lv}(t) = k \cdot \left[ \frac{\left( \frac{t}{\alpha_1 T} \right)^{n_1}}{1 + \left( \frac{t}{\alpha_1 T} \right)^{n_1}} \right] \cdot \left[ \frac{1}{1 + \left( \frac{t}{\alpha_2 T} \right)^{n_2}} \right] \tag{4}$$

The first and second terms in Eq 4 represent ventricle contraction and relaxation, respectively. $k$ is a scaling factor that defines the maximal value of elastance, $T$ is the heart period, $\alpha_1$, $\alpha_2$ are shape parameters, and $n_1, n_2$ control the steepness of the curve.

To account for the mechanical function of the atria, the atrial pressure $P_{la}$ is represented as a linear function of its instantaneous volume $V_{la}$, whose slope $E_{la}$ represents the elastic properties of the atrial wall:

$$P_{la}(V_a, t) = E_{la}(t) \cdot (V_{la}(t) - V_{d,la}), \tag{5}$$

$$E_{la}(t) = E_{la,max} \left( e_{la}(t) + \frac{E_{la,min}}{E_{la,max}} \right) \tag{6}$$

where $e_{la}(t)$ is a Gaussian driving function that cycles between atrial diastole and systole:

$$e_{la}(t) = \exp(-B_{la} \cdot (t - C_{la})^2) \tag{7}$$

Using $B_{la}$ and $C_{la}$, it is possible to control the rise and peak of the atrial systole.

**1.2.3 Systemic and pulmonary circulations.** Concerning the circulatory models [21], the volume change of each cardiac or vessel chamber is calculated from the net flow: $\Delta V(t) = \int (Q_{in} - Q_{out}) dt$. The flows are defined by the pressure gradient across chambers and a resistance: $Q = {}^{\Delta}P/_R$. The pressure of arterial and venous vessels are defined as an elastance dependent linear relationship, similar to Eq 1. The circulatory model allows for the simulation of systolic and diastolic aortic pressures ($P_{ao,sys}^{model}$ and $P_{ao,dias}^{model}$).

**1.2.4 Cardiac valves.** The cardiovascular system (CVS) model was coupled to a detailed representation of the heart valves dynamics (mitral, aortic, tricuspid and pulmonary) according to [22]. The relation between the pressure gradient ($\Delta P$) and the fluid flow ($Q$) across an open valve is approximated by the Bernoulli equation (Eq 8):

$$\Delta P = BQ|Q| + L\frac{dQ}{dt}, \tag{8}$$

$$B = \frac{\rho}{2A_{eff}^2}, \quad L = \frac{\rho l_{eff}}{A_{eff}} \tag{9}$$

where $B$ and $L$ are respectively the Bernoulli resistance and the blood inertance. Parameter $\rho$ stands for the blood density, $A_{eff}$ is the effective cross-sectional area of the valve (Eq 10) and $l_{eff}$ is the effective length of the valve:

$$A_{eff}(t) = [A_{eff,max}(t) - A_{eff,min}(t)]\xi(t) + A_{eff,min}(t) \tag{10}$$

$$\frac{d\xi}{dt} = \begin{cases} (1 - \xi)K_{vo}\Delta P, & if \, \Delta P > 0 \\ \xi K_{vc}\Delta P, & if \, \Delta P \leq 0 \end{cases} \tag{11}$$

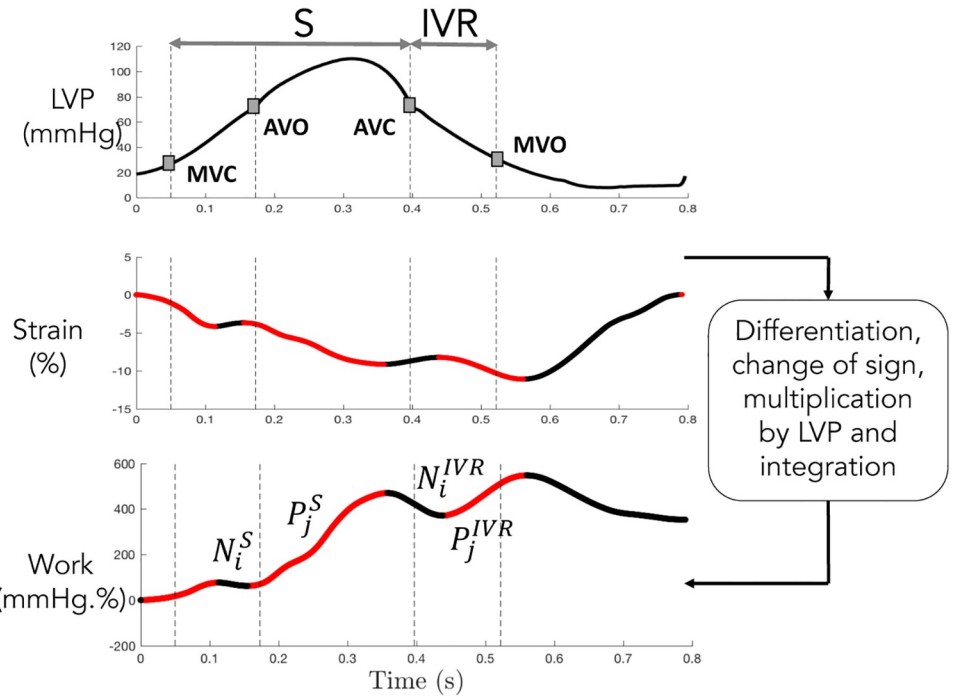

**Fig 2. Calculation of positive and negative segmental work.** Positive ($P_j$) and negative ($N_i$) work are marked respectively as red and black. Phase *S* corresponds to isovolumic contraction and ejection. *IVR* is the isovolumic relaxation. S phase is defined by the time interval spanning from MVC to AVC, whereas the IVR phase is defined between AVC and MVO.

$A_{eff,max}$ and $A_{eff,min}$ are the maximum and minimum effective areas. The rate of opening $\xi$ (*t*) describes the dynamic of the valve position (Eq 11), in response to $\Delta P$. $K_{vo}$ and $K_{vc}$ are the rate coefficients for valve opening and closure, respectively.

### 1.3 Estimation of myocardial work

Segmental myocardial work, as proposed by Russell et al [10], could be calculated from 1) the clinical strain signals, deduced from TTE, and 2) the LV pressure obtained invasively by catheterization ($P_{LV}^{exp}$) or the patient-specific pressure ($P_{LV}^{model}$) from the model-based approach. The instantaneous power was first obtained by multiplying the strain rate, obtained by differentiating the strain curve, and the instantaneous LV pressure. Then, segmental myocardial work was calculated by integrating the power over time, during the cardiac cycle from mitral valve closure until mitral valve opening.

Positive and negative work [12] were determined as the ascending and descending parts of the curves (Fig 2), during isovolumic contraction and ejection (*S* phase) and isovolumic relaxation (*IVR* phase). Then, positive segmental work $W_p$ (respectively $W_n$) is defined as the sum of positive (respectively negative) variations for each segment *k* and for each phase (*S* and *IVR*):

$$W_{p,k}^{S} = \sum_{i} P_{i,k}^{S}, \quad W_{n,k}^{S} = \sum_{j} N_{j,k}^{S}, \tag{12}$$

$$W_{p,k}^{IVR} = \sum_{i} P_{i,k}^{IVR}, \quad W_{n,k}^{IVR} = \sum_{j} N_{j,k}^{IVR} \tag{13}$$

where $P_i$ (respectively $N_j$) is the variation associated with each ascending (respectively descending) parts $i$ (respectively $j$) of the segmental work (Fig 2). The indices $i$ (respectively $j$) are comprised between 1 and the total number of ascending (respectively descending) parts. Finally, global constructive ($GCW$) and wasted ($GWW$) work are defined as mean values over all segments:

$$GCW = \frac{1}{k}\sum_{k=1}^{K}(W_{p,k}^{S} + W_{n,k}^{IVR}), \tag{14}$$

$$GWW = \frac{1}{k}\sum_{k=1}^{K}(W_{n,k}^{S} + W_{p,k}^{IVR}) \tag{15}$$

where $K$ is the total number of segments. $GCW$ represents segmental shortening during the systole, i.e. effective energy for blood ejection, and lengthening during $IVR$, whereas $GWW$ corresponds to segmental stretching during the systole, i.e. energy loss for blood ejection and shortening during the isovolumic relaxation phase. GWE is defined as the global work efficiency:

$$GWE = \frac{GCW}{GCW + GWW} \tag{16}$$

### 1.4 Model-based, patient-specific LV pressure estimation

**1.4.1 Sensitivity analysis.** The objective of the sensitivity analysis is to determine the sets of ventricular $\{\mathbf{X}_{LV}\}$ and circulatory $\{\mathbf{X}_{art}\}$ parameters that have the most important influence on the gradient of pressure ($\Delta P^{model} = max(P_{LV}^{model}) - P_{ao,sys}^{model}$) between LV and aorta. Using the Morris elementary effects method [26], the sensitivity of each parameter is estimated by repeated measurements of a simulation output $Y$ with parameters $\mathbf{X}$, while changing one parameter $X_j$ at a time. The method consists in the generation of several random trajectories through the parameter space; each trajectory being associated with an estimation of the Elementary Effects $EE_j^*$ of a parameter $X_j$ on output $Y$:

$$EE_j^* = \left| \frac{Y(X_1, \ldots, X_j, \ldots) - Y(X_1, \ldots, X_j + \Delta, \ldots)}{\Delta} \right| \tag{17}$$

where $\Delta$ is a predefined variation. For each $X_j$, the mean $\mu_j^*$ and standard deviation $\sigma_j$ of $r$ elementary effects ($EE_j$) are calculated. A large value of $\mu_j^*$ indicates a significant effect of $X_i$ on $Y$, whereas a large $\sigma_j$ value is related to either non-linear or strongly interacting variables. In order to establish a global rank of importance among parameters, we calculated the Euclidean distance $D_j$ in the $\mu^* - \sigma$ plane, from the origin to each $(\mu_j^*, \sigma_j)$ point:

$$D_j = \sqrt{\left(\mu_j^*\right)^2 + \sigma_j^2} \tag{18}$$

being parameters with high sensitivity or strong interactions those presenting the highest values for $D_j$. Analysis were performed with $Y = \Delta P^{model}$ and, for each parameter $X_j$, the range of possible values was defined as ±30% of the initial values (S1 File).

**1.4.2 Parameter identification.** The parameter identification process is included into a Monte-Carlo cross-validation approach (Fig 3). For all patients, the maximum effective area $A_{eff,max}$ parameter was fixed to the observed AVA, measured from TTE. Available data from the 12 patients were divided randomly into two sets of 6 patients (training and test sets). This

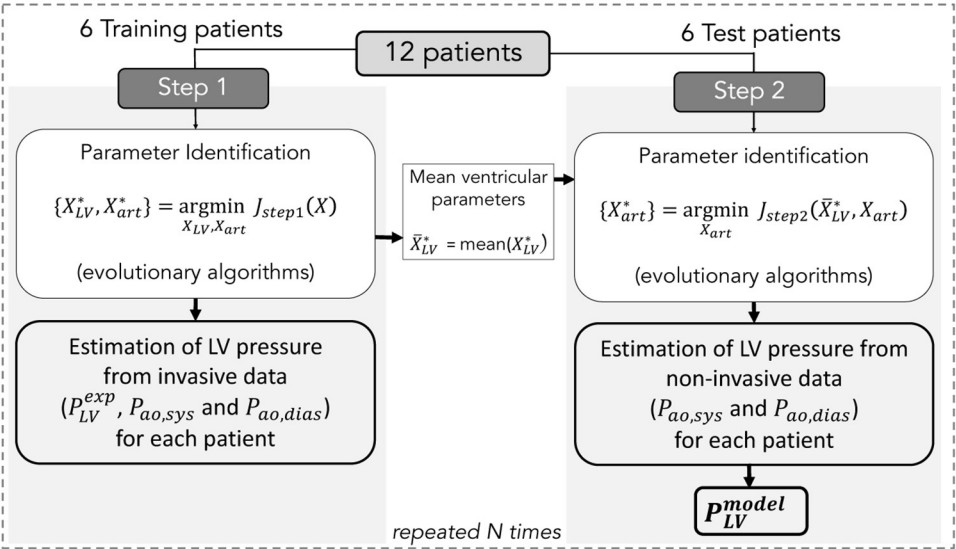

**Fig 3. Two steps of the identification process.** Step 1 consists in the minimization of $J_{step1}$ for the identification of $\{\mathbf{X}_{LV}, \mathbf{X}_{art}\}$ from invasive LV pressure and non-invasive arterial pressure. Step 2 consists in the minimization of $J_{step2}$ for the identification of $\{\mathbf{X}_{art}\}$ from non-invasive arterial pressure. Finally, $P_{LV}^{model}$ is estimated for each patient from $\overline{\mathbf{X}}_{LV}^{*}$ and $\mathbf{X}_{art}^{*}$.

random selection process was applied $N$ times ($n \leqslant N$) and for each realization $n$ a two-step procedure was applied. The following sections provide more details on these steps.

**Step 1**: For each training patient, a parameter identification stage was implemented to find the best set of parameters $\{\mathbf{X}_{LV}^{*}, \mathbf{X}_{art}^{*}\}$ that minimises the error function between simulation outputs and experimental signals:

$$J_{step1} = J_{PLV} + J_{Pao,sys} + J_{Pao,dias} \tag{19}$$

$J_{Pao,sys}$, $J_{Pao,dias}$ and $J_{PLV}$ could be defined as:

$$J_{Pao,sys} = \mid P_{ao,sys}^{exp} - P_{ao,sys}^{model} \mid, \tag{20}$$

$$J_{Pao,dias} = \mid P_{ao,dias}^{exp} - P_{ao,dias}^{model} \mid, \tag{21}$$

$$J_{PLV} = \frac{1}{T_c} \sum_{t_e=0}^{T_c-1} \mid P_{LV}^{exp}(t_e) - P_{LV}^{model}(t_e) \mid \tag{22}$$

where $t_e$ corresponds to the time elapsed since the onset of the identification period and $T_c$ is the duration of a cardiac cycle. The error function $J_{step1}$ was minimised using evolutionary algorithms (EA). These stochastic search methods are founded on theories of natural evolution, such as selection, crossover and mutation [27]. After this identification step, ventricular parameters were fixed equal to the average values over all the training patients ($\overline{\mathbf{X}}_{LV}^{*}$ = mean ($\mathbf{X}_{LV}^{*}$)).

**Step 2**: For each test patient, only circulatory parameters $\{\mathbf{X}_{art}\}$ were identified by minimising the error function:

$$J_{step2} = J_{Pao,sys} + J_{Pao,dias} \tag{23}$$

From the best set of parameters $\{\mathbf{X}_{art}^*\}$, LV pressure $P_{LV}^{model,i}$ was simulated for each test patient and for each iteration $i$ of the 2-step identification algorithm. Then, $GCW^{model,i}$ and $GWW^{model,i}$ were calculated from $P_{LV}^{model,i}$ of each patient. Therefore, at the end of the $N$ iterations, a set of $i$ simulated pressure and work indices was generated for each patient and averaged markers were determined: $GCW^{model} = \overline{GCW^{model,i}}$, $GCW^{model} = \overline{GCW^{model,i}}$ and $GWE^{model} = \overline{GWE^{model,i}}$.

## 1.5 Comparison between simulations and experimental data

**1.5.1 Comparison of estimated and measured pressures.** Inspired form [14], estimated $P_{LV}^{model}$, $P_{ao,sys}^{model}$ and $P_{ao,dias}^{model}$ were compared with measured pressures by calculating the total relative error defined as:

$$e_\% = 50\left(\frac{\parallel P_{LV}^{exp} - P_{LV}^{model} \parallel}{\parallel P_{LV}^{exp} \parallel}\right) + 50\left(\frac{\mid P_{ao,sys}^{exp} - P_{ao,sys}^{model} \mid}{\mid P_{ao,sys}^{exp} \mid} + \frac{\mid P_{ao,dias}^{exp} - P_{ao,dias}^{model} \mid}{\mid P_{ao,dias}^{exp} \mid}\right) \quad (24)$$

where $\parallel.\parallel$ stands for the vectorial 1-norm. A linear regression was performed on all the points from experimental and simulated pressure waveforms. The slope ($\beta$) and coefficient of determination ($R^2$) were deduced from the linear regression.

**1.5.2 Comparison of estimated and measured work indices.** In this paper, $GCW$, $GWW$, and $GWE$ were calculated in two different manners: 1) $GCW^{exp}$, $GWW^{exp}$ and $GWE^{exp}$ using the invasive experimental pressure $P_{LV}^{exp}$, and 2) $GCW^{model}$, $GWW^{model}$ and $GWE^{model}$ using the proposed patient-specific pressure $P_{LV}^{model}$ from the model-based approach. The goodness of work estimations was evaluated by performing a linear regression using indices calculated from invasive experimental and the proposed model-based pressures. Bland-altman (BA) plots were also presented for the three work indices.

# 2 Results

## 2.1 Hemodynamic simulations

Fig 4 illustrates the hemodynamic simulation results of the proposed computational model; the LV and aortic pressures in healthy and AS subjects. Concerning the healthy subject, systolic LV pressure is equal to 120 mmHg, and the aortic pressure varies between 50 and 120 mmHg. AS was represented as a decrease in the $A_{eff,max}$ parameter (from 2.5 to 0.75 $cm^2$). In Fig 4, it is observed an important gradient pressure between LV (0-150 mmHg) and aorta (50-110 mmHg), characteristic of an AS, in which the narrowing of the aortic valve opening evokes an LV pressure overload.

## 2.2 Sensitivity analysis

Sensitivity results evaluated on the gradient of systolic pressure between LV and aorta ($\Delta P^{model}$), are presented in Fig 5, only showing those parameters having the highest sensitivities. Fig 5 shows the 25 most relevant parameters based on their $D_j$ index; $\mu_j^*$ and $\sigma$ are also represented. The most influential parameter corresponds to the effective area of the aortic valve ($A_{eff,max}$). In fact, a decrease of the effective area causes an increase in the ventricular systolic pressure, and consequently, on the gradient of systolic pressure between LV and aorta. Parameters related to the elastance of the LV ($E_{es,lv}$ and $\alpha_2$) have also a high sensitivity on the gradient of systolic pressures. $E_{es,lv}$ corresponds to the maximum LV elastance and is related to myocardial contractility. $\alpha_2$ represents the shape parameter related to the LV relaxation phase.

$A_{eff,max}$ presents the highest sensitivity. Fortunately, this parameter can be non-invasively observed and has been fixed to the AVA value measured from TTE specifically to each patient.

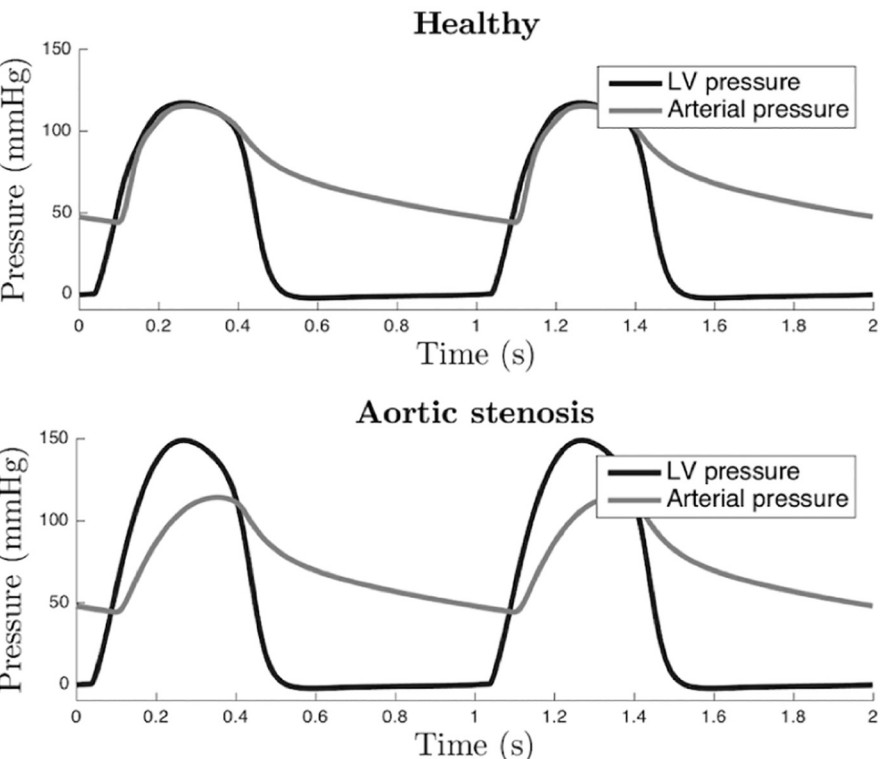

**Fig 4. Example simulated LV and arterial pressure for a healthy (top) and an aortic stenosis subject (bottom).**

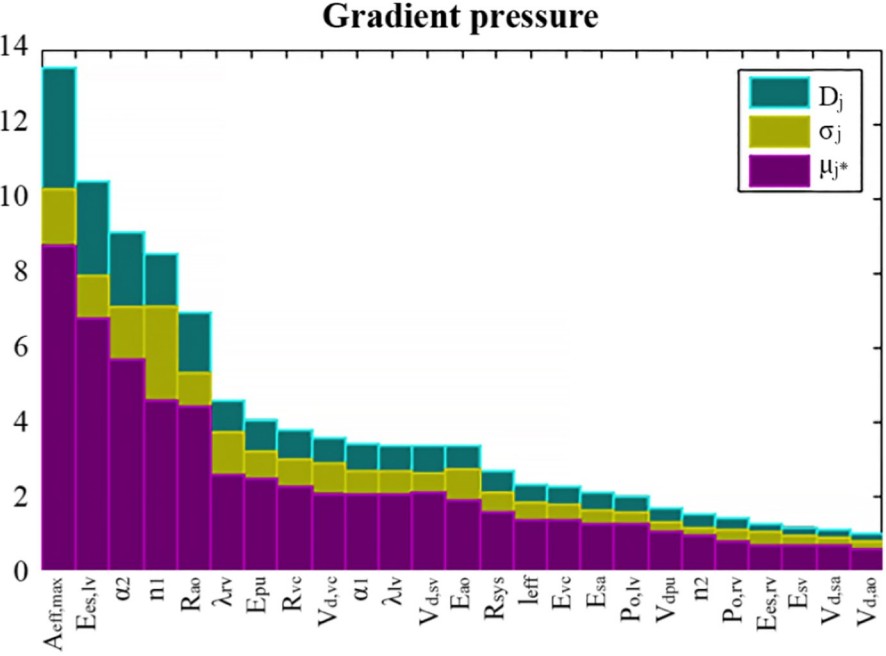

**Fig 5. Most influential parameters on the gradient of systolic pressure between LV and aorta according to Morris sensitivity results.** For each parameter, the distance $D_i$ (green bars), the absolute mean $\mu_i^*$ (purple bars) and the standard deviation $\sigma$ (yellow bars) of the elementary effects are represented.

The ventricular and circulatory parameters with the highest sensitivities were selected for ventricular and circulatory parameter estimations: $\mathbf{X}_{LV} = \{E_{es,lv}, \lambda_{lv}, P_{0,lv}, \alpha_1, \alpha_2, n_1, n_2\}$ and $\mathbf{X}_{art} = \{E_{ao}, E_{vc}, E_{sa}, E_{sv}, Vd_{ao}, Vd_{vc}, Vd_{sa}, Vd_{sv}, R_{ao}, R_{sys}, R_{vc}\}$.

Except for $\{\mathbf{X}_{LV}, \mathbf{X}_{art}\}$, model parameter values were selected from the publications from which each model was originally based on: ventricular and circulatory parameters were taken from [16–18, 23], heart valve parameters were adapted from [22], and cardiac electrical conduction system from [19].

## 2.3 Patient-specific model-based pressure

**2.3.1 Step 1: Estimation of LV pressure from invasive data.**   Concerning step 1 of the parameter identification, there was a good agreement between estimated and measured pressure waveforms (Fig 6). Mean $R^2$ was equal to 0.96 (min: 0.91, max: 0.99). Mean slope and intercept of the regression line between the simulated and the measured pressure data were 1.04 (95% confidence interval: 1.0, 1.09) and -8.48 (-8.52, -8.44) mmHg respectively. Mean total relative error was equal to 11.9% and ranged from 6.4% to 17.3%.

**2.3.2 Step 2: Estimation of LV pressure from systolic and diastolic pressure values.** Concerning step 2 of the parameter identification, LV pressure waveforms (Fig 7) are only estimated from systolic and diastolic pressure values and $A_{eff,max}$ has been fixed to the AVA value measured from TTE specifically to each patient. Slope and intercept of the regression line were 1.03 (0.92, 1.14) and—7.74 (-7.63, -7.85) mmHg respectively, and mean $R^2$ was 0.91. Total relative error ranged between 5.9% and 17.40% and average value is 12.27%.

## 2.4 Comparison of global cardiac work indices

Fig 8 presents scatter and BA plots for GCW, GWW and GWE indices. Correlations between measures and model-based estimations were respectively 0.88 (p < 0.0001) and 0.80 (p < 0.003) for GCW and GWW. When considering both constructive and wasted work

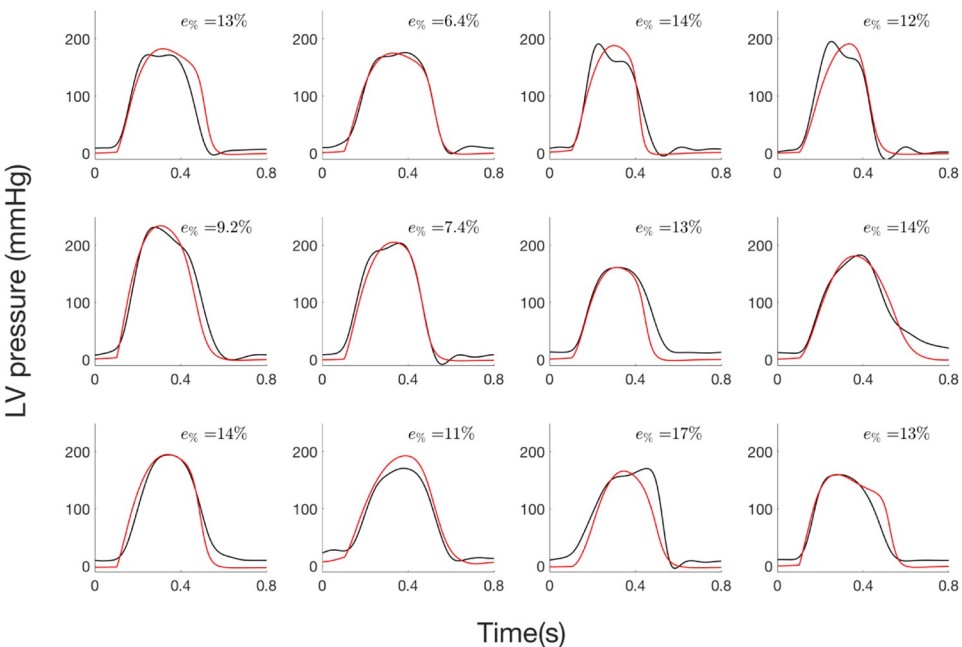

**Fig 6. LV pressure of the 12 AS patients from step 1: i) experimental (black) and ii) simulated curve (red) curves.**

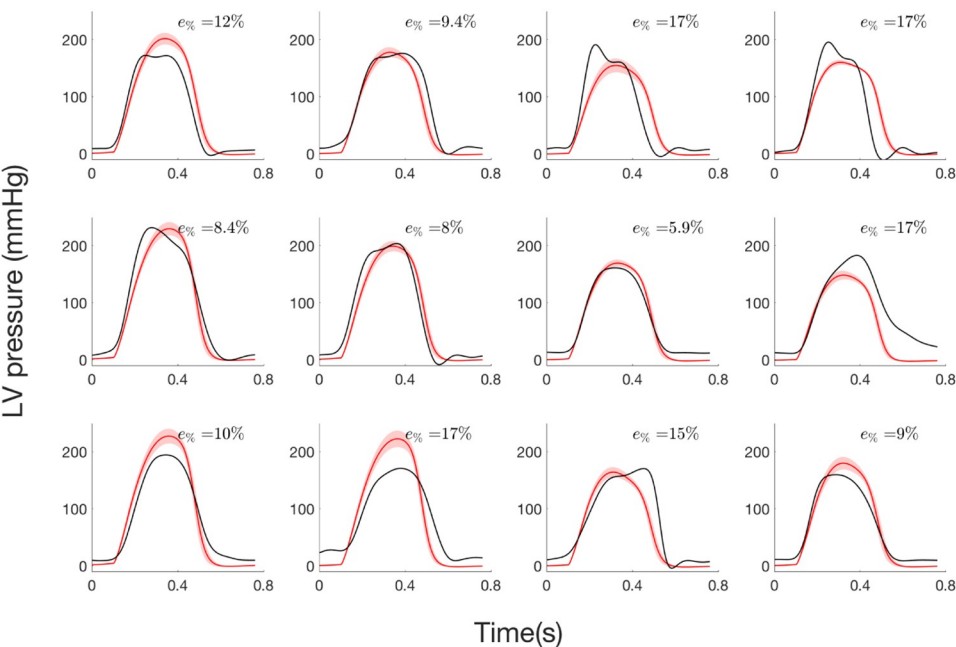

**Fig 7. LV pressure of the 12 AS patients from step 2: i) experimental curve (black), ii) average and standard deviation of simulated curve (red).**

indices, global correlation was equal to 0.96 (p < 0.0001). In BA analysis, mean bias were -140 mmHg.% and -12 mmHg.% respectively for GCW and GWW, which correspond to relative bias equal to 3.47% and 2.93% with respect to maximum GCW and GWW values. For global work efficiency, correlation was 0.91(p < 0.0001) and mean bias was equal to 0. For GWE, one patient is outside the 95% limits of agreement and corresponds to the third patient of the first row on Fig 6 and 7. For this patient, the synchronisation, between peaks of simulated and experimental pressures, is less good, showing the importance of time corresponding to peak pressure for work evaluation.

## 3 Discussion

In this paper, a patient-specific model-based estimation method was proposed in order to evaluate constructive, wasted myocardial work and global work efficiency on patients diagnosed with aortic stenosis. The main contributions of this study concern: *i)* the proposition of an integrated model of cardiovascular system model, *ii)* the analysis of this cardiovascular model in order to select the most sensitive parameters to be identified in a patient-specific manner, *iii)* a parameter identification approach able to reproduce LV pressure specifically to each patient and *iv)* the experimental validation of the proposed method through a cross-validation technique applied on 12 AS patients, in order to quantitatively evaluate GCW, GWW and GWE indices.

The heart valve model, proposed by [22], was coupled to a CVS model that includes representations of cardiac electrical activity, cardiac cavities and the circulation, developed by our group [16–21]. The integrated model is able to predict the influence of valve motion on hemodynamics in both normal and stenosis cases. The sensitivity analysis, performed on the integrated model, highlights the importance of effective area of the aortic valve and parameters related with LV elastance on the pressure gradient between LV and aorta. In fact,

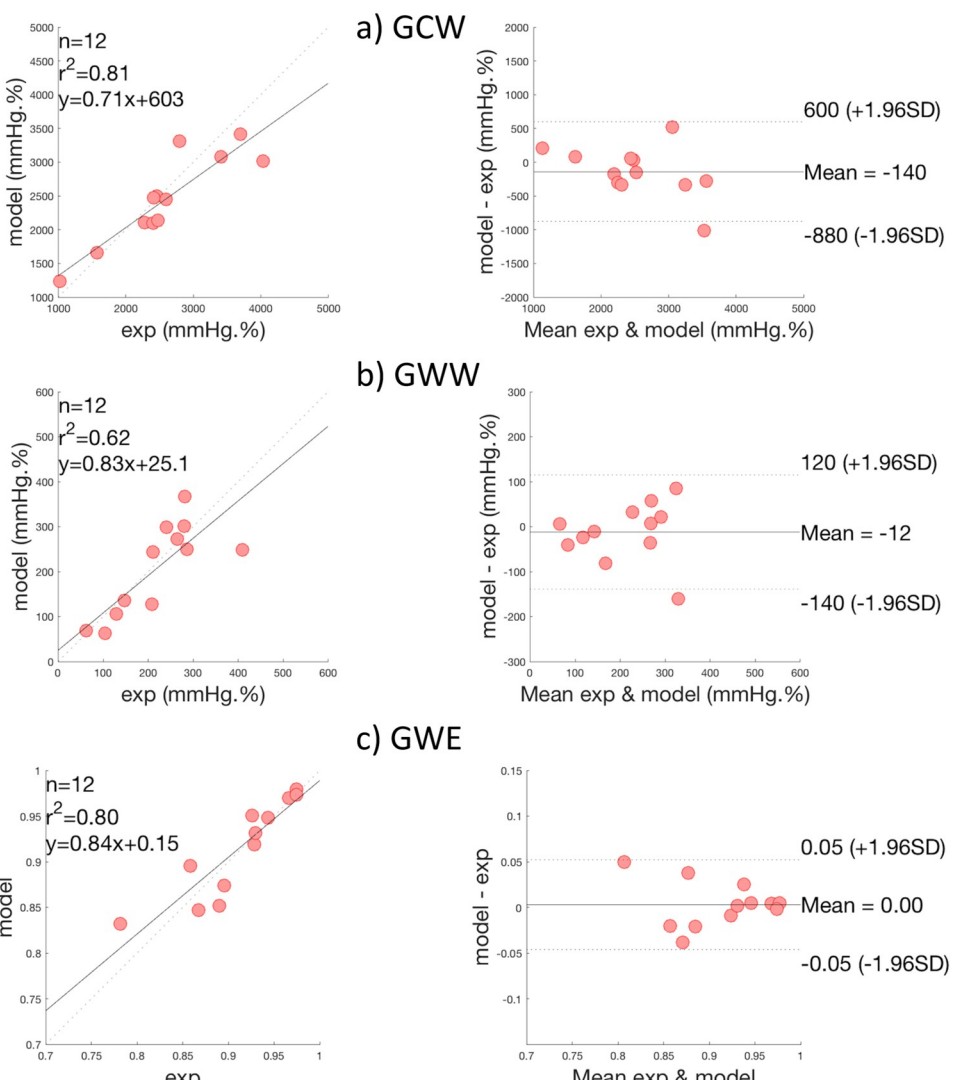

**Fig 8. Results of global work indices comparison, on all patients.** Scatter plots and Bland—Altman analysis of: a) Global Constructive Work (GCW), b) Global Wasted Work (GWW) and c) Global Work Efficiency (GWE).

modifications of valve effective area, observed in stenosis, lead to an increased aortic resistance and to an elevated pressure gradient across the valve [13]. When the blood flows through a narrowed aortic valve, the hemodynamic conditions could also lead to modifications of ventricular elastance [28].

The most influential LV and aortic parameters found after sensitivity analysis were then identified for each one of the 12 patients. One of the main originality of the approach was to apply a Monte-Carlo cross-validation approach for the patient-specific estimations of LV pressures. In order to build the cost function, experimental and simulated pressures were synchronised on QRS peaks of synthesized and experimental ECG. In the first step of the identification process, model parameters were identified from invasive measured LV pressures, as well as systolic and diastolic arterial pressure values. Results show a good agreement between estimated and measured pressure waveforms. Concerning the second step of the identification, only systolic, diastolic arterial pressure values and AVA echocardiography

estimations were used to identify some model parameters and to estimate LV pressure waveform. Although errors slightly increase compared to step 1, the approach has the advantage of using only non-invasive data for the estimation of LV pressure waveforms.

As shown in previous work of our team [12], although LV pressure is imprecise, the estimation of LV work could be accurate. In fact, even errors between model-based and measured pressures are around 12%, mean relative bias in BA analysis were 3.47%, 2.93% and 0.0% respectively for GCW, GWW and GWE. The consistency of LV work estimation could be explained by: i) the temporal integration, which induces a smoothing of the difference between measured and estimated works and ii) relative precision of the estimation of the pressure between AVO and AVC. Although the estimation of the LV pressure is imperfect, the non-invasive estimation of global myocardial work indices obtained from modelling approach strongly correlates with invasive measurements and the proposed estimation of LV myocardial work appears as clinically relevant.

Myocardial work indices are novel tools that have been validated in a variety of pathologies, including the response to cardiac resynchronization therapy (CRT) [29]. In particular, Russell et al. have shown that regional differences in myocardial work have a strong correlation with regional myocardial glucose metabolism, as evaluated using PET imaging [9]. However, the assessment of constructive and wasted work, in the case of AS is difficult because the estimation of peak LV pressure is complicated without any invasive measurement. To our knowledge, this paper presents the first method for the estimation of myocardial work, based on a physiological model, rather than a template-based estimate, such as in [9]. In this case, the model-based method allows for the integration of physiological knowledge in the evaluation of myocardial work indices. *In silico* assessment of clinical parameters, specifically to each patient, has the advantage of taking into account characteristics associated with the subject and pathology. For instance, by integrating a representation of the pathophysiology of the aortic valves within this physiological model, it becomes adapted to the case of aortic stenosis.

Results show globally a good agreement between work index estimations from LV pressure obtained with patient-specific simulations and with experimental measurements. The evaluation of cardiac work, in the case of AS, is promising because it could be a simple and physiological alternative to more complex and costly investigations (cardiac MRI,..) for the evaluation of myocardial contractility and residual myocardial viability [30]. The assessment of regional myocardial work might be particularly important for the prognosis of patients with severe asymptomatic AS without LV dysfunction. In fact, the timing and indications for surgical intervention in this population remain controversial as the aortic-valve replacement is not recommended despite in the increased risk of cardiovascular mortality [31]. Indeed, as LVEF remains imperfect in asymptomatic AS to determine the optimal delay for the surgery, global longitudinal strain appears to have a better prognostic significance [32] and we can suppose that myocardial work will be a robust complementary index independent of afterload condition. In fact, because afterload data are included in the calculation of myocardial work in the form of LV pressure, the assessment of myocardial work might represent a more robust parameter with respect to the assessment of LV strain or other strain-derived parameters [33]. Although it will be important to confirm these assumptions and to validate the approach in a cohort of AS patients, this paper is a first essential step for the proposition of work estimation based on computational modelling. The proposed methodology should be evaluated on a larger prospective clinical database in the future and we believe that model-based work indices, especially GWE, could be promising to improve the assessment of LV mechanical efficiency in AS.

One limitation of this work concerns the number of patients included in this study. Although it appears to be low, it is necessary to recognise that measurement of invasive LV

pressure is particularly difficult in AS. In fact, current guidelines discourage catheterization measurement techniques in AS before aortic valve replacement [34]. In this case, catheterization was realised for clinical reasons and all patients were informed. Another limitation is related to the estimation of LV filling pressure, which is not precisely estimated in step 2. In fact, myocardial work is considered in the period from mitral valve closure to mitral valve opening, so inaccuracies before mitral valve closure and after mitral valve opening has no impact on the results [12].

## 4 Conclusion

In this work, we propose an original model-based approach to assess constructive and wasted work in AS patients. The global method is based on a novel approach introducing: *i)* a physiological model of the cardiovascular system, including heart valves and *ii)* a 2-step identification procedure, based on a monte-carlo cross-validation method. The proposed model-based approach was evaluated with data from 12 AS patients for which LV pressure data was acquired invasively. Results show a close match between experimental and simulated LV and aortic pressures. The model-based approach is especially efficient for the evaluation of LV pressure from non-invasive data (systolic, diastolic pressures and aortic valve area). Moreover, estimations of constructive, wasted work and global work efficiency were consistent with indices calculated from measured experimental pressures, showing the model ability to produce realistic LV pressure for the calculation of work indices.

More extensive evaluations including a greater population of patients, as well as the analysis on a prospective study should be performed in the future. Furthermore, the proposed model could be enriched by including a regional description of myocardium [20]. Nevertheless, this paper presents the first model-based approach towards the evaluation of myocardial work indices in AS patients and, thus, provide a step forward the characterisation of the complex LV mechanics of patients with AS.

## Supporting information

**S1 Data.**
(XLSX)

**S2 Data.**
(XLSX)

**S1 File.**
(PDF)

## Author Contributions

**Conceptualization:** Arnaud Hubert, Elena Galli, Erwan Donal, Alfredo I. Hernández, Virginie Le Rolle.

**Data curation:** Kimi P. Owashi, Arnaud Hubert, Elena Galli, Virginie Le Rolle.

**Formal analysis:** Kimi P. Owashi, Alfredo I. Hernández, Virginie Le Rolle.

**Funding acquisition:** Alfredo I. Hernández, Virginie Le Rolle.

**Investigation:** Kimi P. Owashi, Arnaud Hubert, Erwan Donal, Alfredo I. Hernández, Virginie Le Rolle.

**Methodology:** Kimi P. Owashi, Elena Galli, Alfredo I. Hernández, Virginie Le Rolle.

**Project administration:** Alfredo I. Hernández, Virginie Le Rolle.

**Software:** Kimi P. Owashi, Virginie Le Rolle.

**Supervision:** Erwan Donal, Alfredo I. Hernández, Virginie Le Rolle.

**Validation:** Kimi P. Owashi, Arnaud Hubert, Elena Galli, Alfredo I. Hernández, Virginie Le Rolle.

**Visualization:** Kimi P. Owashi, Arnaud Hubert, Elena Galli, Alfredo I. Hernández, Virginie Le Rolle.

**Writing – original draft:** Kimi P. Owashi, Alfredo I. Hernández, Virginie Le Rolle.

**Writing – review & editing:** Kimi P. Owashi, Arnaud Hubert, Elena Galli, Erwan Donal, Alfredo I. Hernández, Virginie Le Rolle.

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
