## [Decision Letter · Decision Letter 0]

10 Dec 2019

PONE-D-19-26305

Model-based estimation of left ventricular pressure and myocardial work in aortic stenosis

PLOS ONE

Dear Dr Le Rolle,

Thank you for submitting your manuscript to PLOS ONE. After careful consideration, we feel that it has merit but does not fully meet PLOS ONE’s publication criteria as it currently stands. Therefore, we invite you to submit a revised version of the manuscript that addresses the points raised during the review process.

You need to address all comments of the two reviewers. You should be able to provide convincing data on the potential usefulness of your model as compared to current clinical evaluation. 

We would appreciate receiving your revised manuscript by Jan 24 2020 11:59PM. To enhance the reproducibility of your results, we recommend that if applicable you deposit your laboratory protocols in protocols.io, where a protocol can be assigned its own identifier (DOI) such that it can be cited independently in the future. For instructions see: http://journals.plos.org/plosone/s/submission-guidelines#loc-laboratory-protocols

We look forward to receiving your revised manuscript.

Kind regards,

Cécile Oury

Academic Editor

PLOS ONE

Journal Requirements:

2. Please provide additional details regarding participant consent.

In the ethics statement in the Methods and online submission information, please ensure that you have specified (i) whether consent was informed and (ii) what type you obtained (for instance, written or verbal, and if verbal, how it was documented and witnessed).

If the need for consent was waived by the ethics committee, please include this information.

3. Please ensure that you refer to Figure 7 in your text as, if accepted, production will need this reference to link the reader to the figure.

Reviewers' comments:

Reviewer's Responses to Questions

**Comments to the Author**

1. Is the manuscript technically sound, and do the data support the conclusions?

Reviewer #1: Yes

Reviewer #2: Partly

2. Has the statistical analysis been performed appropriately and rigorously? 

Reviewer #1: Yes

Reviewer #2: Yes

3. Have the authors made all data underlying the findings in their manuscript fully available?

Reviewer #1: Yes

Reviewer #2: Yes

4. Is the manuscript presented in an intelligible fashion and written in standard English?

Reviewer #1: Yes

Reviewer #2: Yes

5. Review Comments to the Author

Reviewer #1: In this paper, Owashi et al. performed a model-based estimation of LV pressure and myocardial work that can be applied in the context of patients with aortic stenosis (AS). They validated their model in a cohort of 12 patients with AS (11 severe and 1 moderate AS). Authors reported model errors around 12% for the estimation of LV pressure.

I have few comments for the authors regarding the present manuscript.

1 – Authors reported a good correlation between estimated and measured pressure (end of page 10). However, even if the correlation coefficient was relatively high (>0.90), the intercept of the slope pointed out an underestimation ranged between 8 to 9 mmHg. Similar findings are identified in the second step of the analysis. Was this difference between model-based estimation and measured pressure clinically relevant? What was the clinical significance of this “systematic” underestimation? How can we ascertain the low clinical impact of this kind of LV pressure underestimation, especially in a context of AS?

2 – Could you validate your estimation of myocardial work based on clinical gold standard currently used in clinic?

3 – What is the additive value of the estimation of myocardial pressure or work as compared to echo based data including AVA and LV function parameters? A validation of the clinical impact of this approach will considerably strengthen the paper.

4 – What is the efficacy of the model in patients with a more advance stage of the LV dysfunction?

5 – Authors provided echo data for each patient in a supplemental table. The LVEDV/LVESV should be carefully reviewed by the authors. How did you calculate LVEF? SV and strain data could be of interest.

Reviewer #2: This paper describes a model-based approach for estimating left ventricular constructive and wasted work, and work efficiency, in patients with aortic stenosis. The model is very well suited to the question and the results are quite impressive. The main limitation is the small sample size of only 12 patients; it is therefore difficult to tell how robust the results are.

1. Line 122. The equation for calculating volume is missing a reference volume. The integral of flow can only be used to calculate a volume change, not an absolute volume.

2. I assume Section 1.3 relates to the clinical data. Please make this clear at the start of this section, as it follows the model description.

3. How were the S and IVR periods of the cardiac cycle defined?

4. Line 170-171. “For each parameter Xj, a number r of elementary effects are calculated to estimate the mean (μ∗) and standard deviation (σ) of the effects.” I don’t quite follow this statement. Do the authors mean that a range of Xj values are tested and then the average effect (and SD) is taken as a measure of sensitivity? If so, how was the range of Xj values determined? And was Xj adjusted up and down? If so taking the mean would be inappropriate.

5. Figure 4 specifies a) and b), but those labels are not present. Use of (top) and (bottom) is sufficient.

6. In Section 2.3 it is unclear why R^2 values are reported, as these are unreliable measures of agreement. E.g. High R^2 can be obtained even when there is poor agreement, as in the two bottom left panels of Figure 7. e% values are better. I may have missed it, but what is beta? It would be helpful to remind readers of the error measures in the Figure legend.

7. In Section 2.4, it may be helpful to also explain the biases in relative terms, as I (and perhaps other readers) don’t have a feel for whether -140 mmHg.% is a big or small number.

8. Line 272. “For GWE, all patients are within the 95% limits of agreement whereas, for GCW and GWW, one patient is outside”. This statement is tautological, as by definition the 95% limits of agreement will always contain all or almost all of the data for a sample of this size. The authors may want to consider whether the one data point could be defined as an outlier, but this would be difficult to judge from such a small sample.

9. Line 274-276. Please check whether this text does indeed refer to the third patient in the first column of Figs 6 and 7, as the timing of the peaks for this case seem to be pretty well aligned, more so than many of the others.

10. Line 334. “we can suppose that myocardial works will be robust complementary indices independ[e]nt of afterload condition.” This may be so, but I would suggest the authors discuss what needs to be done to establish evidence around this question.

11. Line 342. Replace “it is necessary to precise that” with “it is necessary to recognise that”.

12. Line 347-350. It is surprising that filling pressure has no impact on myocardial work. Preload should have a large impact on developed pressure and cardiac output, hence I would have expected a significant sensitivity to this. Can the authors please clarify and explain this?

13. Check references. A number of references have question marks (?) in various places.

Minor:

14. Line 255. change “was equal 11.9% to” to “was equal to 11.9%”

15. Line 298. Change to “the most influential”

16. Line 326. Replace “indice” with “index”

17. Throughout the manuscript, replace “works” with “work”. E.g. “constructive and wasted myocardial work” is correct English.

6. PLOS authors have the option to publish the peer review history of their article (what does this mean?). If published, this will include your full peer review and any attached files.

Reviewer #1: No

Reviewer #2: No

---

## [Author Response · Author response to Decision Letter 0]

27 Dec 2019

Reviewer #1: 

In this paper, Owashi et al. performed a model-based estimation of LV pressure and myocardial work that can be applied in the context of patients with aortic stenosis (AS). They validated their model in a cohort of 12 patients with AS (11 severe and 1 moderate AS). Authors reported model errors around 12% for the estimation of LV pressure.

I have few comments for the authors regarding the present manuscript.

1 – Authors reported a good correlation between estimated and measured pressure (end of page 10). However, even if the correlation coefficient was relatively high (>0.90), the intercept of the slope pointed out an underestimation ranged between 8 to 9 mmHg. Similar findings are identified in the second step of the analysis. Was this difference between model-based estimation and measured pressure clinically relevant? What was the clinical significance of this “systematic” underestimation? How can we ascertain the low clinical impact of this kind of LV pressure underestimation, especially in a context of AS?

We thank reviewer 1 for this interesting question. As shown in previous work of our team (Hubert 2018), although LV pressure is imprecise, the estimation of LV work could be accurate. In fact, even errors between model-based and measured pressures are around 12%, mean bias in BA analysis were -140 mmHg.% and -12 mmHg.% respectively for GCW and GWW, which correspond to relative bias equal to 3.47 % and 2.93 % with respect to maximum GCW and GWW. Mean bias for GWE is equal to 0. The consistency of LV work estimation could be explained by: i) the temporal integration, which induces a smoothing of the difference between measured and estimated works and ii) relative precision of the estimation of the pressure between AVO and AVC. Although the estimation of the LV pressure is imperfect, the non-invasive estimation of global myocardial work indices obtained from modelling approach strongly correlates with invasive measurements and we believe that the proposed estimation of LV myocardial work is clinically relevant. Discussion section was modified in order to better consider these aspects. 

Hubert A et al. Estimation of myocardial work from pressure-strain loops analysis: an experimental evaluation. Eur Heart J Cardiovasc Imaging. 2018;doi:doi: 10.1093/ehjci/jey024. 

2 – Could you validate your estimation of myocardial work based on clinical gold standard currently used in clinic?

The gold standard consists in measuring invasively left ventricular myocardial deformation by sonomicrometry simultaneously with left ventricular pressure by micromanometer. In our study, LV pressure was measured because of clinically utility but it would be not ethical to realize sonomicrometry in patients with aortic stenosis for an observational study. Unfortunately, this gold standard is impossible in human, only in animals, as Urheim et al (2005) did for normal LV. 

Urheim S et al. Regional myocardial work by strain Doppler echocardiography and LV pressure: a new method for quantifying myocardial function. American Journal of Physiology - Heart and Circulatory Physiology. 2005;288:H2375–H2380. 

3 – What is the additive value of the estimation of myocardial pressure or work as compared to echo based data including AVA and LV function parameters? A validation of the clinical impact of this approach will considerably strengthen the paper.

Timing and indications for surgical intervention in patients with severe asymptomatic AS without LV dysfunction remains highly controversial. In fact, surgery is currently not recommended despite in the increased risk of cardiovascular mortality in this population (Kang DH et al., 2019). Global longitudinal strain appears to have a better significance than left ventricular ejection fraction to predict events in this population (Magne J et al., 2019). But it’s well known that GLS is afterload dependant so our hypothesis is that myocardial work, taking in count LV afterload, will better stratify pre-operative and post-operative risk in this population. In fact, because afterload data are included in the calculation of myocardial work in the form of LV pressure, the assessment of myocardial work might represent a more robust parameter with respect to the assessment of LV strain or other strain-derived parameters. 

Discussion was modified in order the better describe the clinical objectives of this approach. 

Kang DH et al. Early surgery or conservative care for asymptomatic aortic stenosis. NEJM 2019, DOI: 10.1056/NEJMoa1912846

Magne J et al. Distribution and Prognostic Significance of Left Ventricular Global Longitudinal Strain in Asymptomatic Significant Aortic Stenosis: An Individual Participant Data Meta-Analysis. JACC Cardiovasc Imaging. 2019;12:84–92

Galli, E, Leclercq, C, Fournet, M, Hubert, A, Bernard, A, Smiseth, OA, Mabo, P, Samset, E, Hernandez, A, Donal, E (2018). Value of Myocardial Work Estimation in the Prediction of Response to Cardiac Resynchronization Therapy. J Am Soc Echocardiogr, 31, 2:220-230.

4 – What is the efficacy of the model in patients with a more advance stage of the LV dysfunction?

For this first approach of the model, we decided to have a homogeneous population so aortic stenosis with normal ejection fraction. But this question is very interesting and should be adressed in further studies.

5 – Authors provided echo data for each patient in a supplemental table. The LVEDV/LVESV should be carefully reviewed by the authors. How did you calculate LVEF? SV and strain data could be of interest.

Indeed, there was an inversion between LVEDV and LVESV in this table. Left ventricular ejection fraction was calculated by manual Simpson’s biplane, realized by a certified cardiologist and followed current recommendations (Lang RM et al., 2015). The data table (data_PlosOne.xls) was modified in the supporting information. 

Lang RM et al. Recommendations for Cardiac Chamber Quantification by Echocardiography in Adults: An Update from the American Society of Echocardiography and the European Association of Cardiovascular Imaging. Eur Heart J Cardiovasc Imaging. 2015;16:233–271

Reviewer #2: 

This paper describes a model-based approach for estimating left ventricular constructive and wasted work, and work efficiency, in patients with aortic stenosis. The model is very well suited to the question and the results are quite impressive. The main limitation is the small sample size of only 12 patients; it is therefore difficult to tell how robust the results are.

We would like to thank the reviewer and we have modified the manuscript in order to improve the article.

Concerning the small sample size, the low number of patients is due to the difficulty of measuring LV pressure in aortic stenosis because current guidelines discourage catheterization measurement techniques. In our study, LV pressure was measured because of clinically utility and all patients were informed.

1. Line 122. The equation for calculating volume is missing a reference volume. The integral of flow can only be used to calculate a volume change, not an absolute volume.

The reviewer is right. The equation describes de flow change: . Section 1.2.3 was corrected. 

\f

2. I assume Section 1.3 relates to the clinical data. Please make this clear at the start of this section, as it follows the model description.

Section 1.3 was modified in order to improve the description of work indices calculated from clinical and simulated data: Segmental myocardial work, as proposed by Russell et al (Russell 2013), could be calculated from 1) the clinical strain signals, deduced from TTE, and 2) the LV pressure obtained invasively by catheterization or the patient-specific pressure from the model-based approach.

Russell K, Eriksen M, Aaberge L, Wilhelmsen N, Skulstad H, Gjesdal O, et al. Assessment of wasted myocardial work: a novel method to quantify energy loss due to uncoordinated left ventricular contractions. Am J Physiol Heart Circ Physiol. 2013;305:H996–H1003,. 

3. How were the S and IVR periods of the cardiac cycle defined?

The definition of S and IVR period were integrated in the legend of figure 2. S phase is defined by the time interval spanning from mitral valve closure (MVC) to aortic valve closure (AVC), whereas the IVR phase is defined between aortic valve closure and mitral valve opening (MVO).

4. Line 170-171. “For each parameter Xj, a number r of elementary effects are calculated to estimate the mean (μ∗) and standard deviation (σ) of the effects.” I don’t quite follow this statement. Do the authors mean that a range of Xj values are tested and then the average effect (and SD) is taken as a measure of sensitivity? If so, how was the range of Xj values determined? And was Xj adjusted up and down? If so taking the mean would be inappropriate.

The method consists in the generation of several random trajectories through the parameter space; each trajectory being associated with an estimation of the Elementary Effects EE∗j of a parameter Xj on output Y : 

where ∆ is a predefined variation. For each Xj , the mean μ∗j and standard deviation σj of r elementary effects (EEj) are calculated. A large value of μ∗j indicates a significant effect of Xi on Y , whereas a large σj value is related to either non-linear or strongly interacting variables. In order to establish a global rank of importance among parameters, we calculated the Euclidean distance Dj in the μ∗ − σ plane, from the origin to each (μ∗j , σj ) point. For each parameter Xj, the range of possible values was defined as +/- 30% of the initial value (supplementary materials). 

Section 1.4.1 was modified.

5. Figure 4 specifies a) and b), but those labels are not present. Use of (top) and (bottom) is sufficient.

Legend of figure 4 was corrected. 

6. In Section 2.3 it is unclear why R^2 values are reported, as these are unreliable measures of agreement. E.g. High R^2 can be obtained even when there is poor agreement, as in the two bottom left panels of Figure 7. e% values are better. I may have missed it, but what is beta? It would be helpful to remind readers of the error measures in the Figure legend.

The goodness of fit was evaluated by performing a linear regression using all the data points from the pressure waveforms, and the coefficient of determination (R2), as well as the slope (), were determined. The R2 and values were removed from figure 6 and 7 and only e% values appear in these figures. Results are presented in sections 2.3.1 and 2.3.2. 

7. In Section 2.4, it may be helpful to also explain the biases in relative terms, as I (and perhaps other readers) don’t have a feel for whether -140 mmHg.% is a big or small number.

In BA analysis, mean bias were -140 mmHg.% and -12 mmHg.% respectively for GCW and GWW, which correspond to relative bias equal to 3.47 % and 2.93 % with respect to maximum GCW and GWW values. Section 2.4 was modified to explain the biases in relative terms. 

8. Line 272. “For GWE, all patients are within the 95% limits of agreement whereas, for GCW and GWW, one patient is outside”. This statement is tautological, as by definition the 95% limits of agreement will always contain all or almost all of the data for a sample of this size. The authors may want to consider whether the one data point could be defined as an outlier, but this would be difficult to judge from such a small sample.

The reviewer is right. This sentence was removed from the manuscript. 

9. Line 274-276. Please check whether this text does indeed refer to the third patient in the first column of Figs 6 and 7, as the timing of the peaks for this case seem to be pretty well aligned, more so than many of the others.

In fact, there was a mistake in this sentence, as it is the third patient in the first row of Figs 6 and 7. Section 2.4 was corrected. 

10. Line 334. “we can suppose that myocardial works will be robust complementary indices independ[e]nt of afterload condition.” This may be so, but I would suggest the authors discuss what needs to be done to establish evidence around this question.

Concerning the influence of loading conditions, it is recognized that the estimation of myocardial work provides an automatic analysis of myocardial performance that is independent of LV afterload (Galli, 2018). Because afterload data are included in the calculation of myocardial work in the form of LV pressure, the assessment of myocardial work might represent a more robust parameter with respect to the assessment of LV strain or other strain-derived parameters. The discussion section was modified.

Galli, E, Leclercq, C, Fournet, M, Hubert, A, Bernard, A, Smiseth, OA, Mabo, P, Samset, E, Hernandez, A, Donal, E (2018). Value of Myocardial Work Estimation in the Prediction of Response to Cardiac Resynchronization Therapy. J Am Soc Echocardiogr, 31, 2:220-230.

11. Line 342. Replace “it is necessary to precise that” with “it is necessary to recognise that”.

The sentence was modified in the manuscript. 

12. Line 347-350. It is surprising that filling pressure has no impact on myocardial work. Preload should have a large impact on developed pressure and cardiac output, hence I would have expected a significant sensitivity to this. Can the authors please clarify and explain this?

We agree with the reviewer. In fact, this sentence is ambiguous and the discussion section was modified. Despite the estimation of filling pressure remains imperfect, myocardial work is considered in the period from MVC to MVO, so inaccuracies before mitral valve closure and after mitral valve opening has no impact on the results. 

13. Check references. A number of references have question marks (?) in various places.

The references section was corrected. 

Minor:

14. Line 255. change “was equal 11.9% to” to “was equal to 11.9%” 

15. Line 298. Change to “the most influential” 

16. Line 326. Replace “indice” with “index” 

17. Throughout the manuscript, replace “works” with “work”. E.g. “constructive and wasted myocardial work” is correct English. 

We thank the reviewer for these suggestions. We have integrated all these corrections in the manuscript.

---

## [Decision Letter · Decision Letter 1]

27 Jan 2020

PONE-D-19-26305R1

Model-based estimation of left ventricular pressure and myocardial work in aortic stenosis

PLOS ONE

Dear Dr Le Rolle,

Thank you for submitting your manuscript to PLOS ONE. After careful consideration, we feel that it has merit but does not fully meet PLOS ONE’s publication criteria as it currently stands. Therefore, we invite you to submit a revised version of the manuscript that addresses the points raised during the review process.

You should answer to the remaining minor comment of reviewer 1.

We would appreciate receiving your revised manuscript within 3 weeks. To enhance the reproducibility of your results, we recommend that if applicable you deposit your laboratory protocols in protocols.io, where a protocol can be assigned its own identifier (DOI) such that it can be cited independently in the future. For instructions see: http://journals.plos.org/plosone/s/submission-guidelines#loc-laboratory-protocols

We look forward to receiving your revised manuscript.

Kind regards,

Cécile Oury

Academic Editor

PLOS ONE

Reviewers' comments:

Reviewer's Responses to Questions

**Comments to the Author**

1. If the authors have adequately addressed your comments raised in a previous round of review and you feel that this manuscript is now acceptable for publication, you may indicate that here to bypass the “Comments to the Author” section, enter your conflict of interest statement in the “Confidential to Editor” section, and submit your "Accept" recommendation.

Reviewer #1: (No Response)

Reviewer #2: All comments have been addressed

2. Is the manuscript technically sound, and do the data support the conclusions?

Reviewer #1: Yes

Reviewer #2: Yes

3. Has the statistical analysis been performed appropriately and rigorously? 

Reviewer #1: Yes

Reviewer #2: Yes

4. Have the authors made all data underlying the findings in their manuscript fully available?

Reviewer #1: Yes

Reviewer #2: Yes

5. Is the manuscript presented in an intelligible fashion and written in standard English?

Reviewer #1: Yes

Reviewer #2: Yes

6. Review Comments to the Author

Reviewer #1: Almost all the comments have been addressed. The remaining point that authors should, at least, acknowledge is lack of validation of the additive predictive value of the estimation of myocardial work (comment #3). Authors provided in their response and the revised manuscript, several hypotheses that could support the usefulness of the estimation of myocardial work, but this should be validate in a cohort of AS patients. Do you have access to an external cohort of AS patients in which myocardial work could be estimated and then in which the additive predicting value could be tested? This would significantly re-inforce the message of the paper. At least, this should clearly be acknowledged by the authors.

Reviewer #2: The authors have responded satisfactorily to all points. I congratulate the authors for an excellent paper.

7. PLOS authors have the option to publish the peer review history of their article (what does this mean?). If published, this will include your full peer review and any attached files.

Reviewer #1: No

Reviewer #2: No

---

## [Author Response · Author response to Decision Letter 1]

3 Feb 2020

Reviewer #1: 

Almost all the comments have been addressed. The remaining point that authors should, at least, acknowledge is lack of validation of the additive predictive value of the estimation of myocardial work (comment #3). Authors provided in their response and the revised manuscript, several hypotheses that could support the usefulness of the estimation of myocardial work, but this should be validate in a cohort of AS patients. Do you have access to an external cohort of AS patients in which myocardial work could be estimated and then in which the additive predicting value could be tested? This would significantly re-inforce the message of the paper. At least, this should clearly be acknowledged by the authors.

We thank reviewer 1 for his comments. The objectives of this paper are 1) to propose a novel tool to estimate non-invasively myocardial work and 2) to compare model-based and invasive indices. This article does not claim to prove the additive predicting value of myocardial work estimation in aortic stenosis. The reviewer is right because this would significantly re-inforce the message of the paper. Unfortunately, we don’t have currently access to any database of AS patients that incudes both echocardiographic strains and invasive ventricular pressures. We are working on the definition of a prospective study for the validation of proposed indices on AS patients and this article is the first step of our approach. We also believe that it is important to clarify the objectives of the paper and to acknowledge that our article does not claim to validate the estimation of myocardial work in a cohort of AS patients. We add several sentences in the introduction and the discussion in order to clarify these aspects.

---

## [Editor Report · Decision Letter 2]

11 Feb 2020

Model-based estimation of left ventricular pressure and myocardial work in aortic stenosis

PONE-D-19-26305R2

Dear Dr. Le Rolle,

We are pleased to inform you that your manuscript has been judged scientifically suitable for publication and will be formally accepted for publication once it complies with all outstanding technical requirements.

With kind regards,

Cécile Oury

Academic Editor

PLOS ONE
---

## [Editor Report · Acceptance letter]

14 Feb 2020

PONE-D-19-26305R2 

Model-based estimation of left ventricular pressure and myocardial work in aortic stenosis 

Dear Dr. Le Rolle:

I am pleased to inform you that your manuscript has been deemed suitable for publication in PLOS ONE. Congratulations! Your manuscript is now with our production department. 

With kind regards,

on behalf of

Dr. Cécile Oury 

Academic Editor

PLOS ONE